# Comparison of three short-course rifamycin-based regimens for the prevention of tuberculosis in patients with end-stage kidney disease: Study protocol for a randomised clinical trial (RIFAKiD-TB trial)

Miguel Santin[1,2]*, Sandra Perez-Recio[1], Maria D. Grijota[1,3], Luis Anibarro[4], Jose M. Barcala[5], Maria L. De Souza-Galvao[6], Paloma Gijon[7], Rafael Luque[8], Francesca Sanchez[9], on behalf of the RIFAKiD team trial[¶]

1 Tuberculosis Unit, Service of Infectious Diseases, Bellvitge University Hospital-Bellvitge Institute for Biomedical Research (IDIBELL), L'Hospitalet de Llobregat, Barcelona, Spain, 2 Department of Clinical Sciences, University of Barcelona, L'Hospitalet de Llobregat, Barcelona, Spain, 3 Department of Fundamental and Medical-Surgical Nursing, University of Barcelona, L'Hospitalet de Llobregat, Barcelona, Spain, 4 Tuberculosis Unit, Department of Internal Medicine, Complexo Hospitalario Universitario de Pontevedra, Pontevedra, Spain, 5 Infectious Diseases Unit, Hospital Universitario de Jerez, Jerez de la Frontera, Cádiz, Spain, 6 Servei de Pneumologia, Hospital Universitari Vall d'Hebron, Barcelona, Spain, 7 Servicio de Microbiología Clínica y Enfermedades Infecciosas, Hospital General Universitario Gregorio Marañón. Instituto de Investigación Sanitaria Gregorio Marañón, Madrid, Spain, 8 Clinical Unit of Infectious Diseases, Microbiology and Preventive Medicine Infectious Diseases Research Group, University of Seville/CSIC/University Virgen del Rocío and Virgen Macarena (IBIS), Sevilla, Spain, 9 Unitat Clínica de Tuberculosis, Hospital del Mar, Barcelona, Spain

¶ Membership of the RIFAKiD Team Trial Group is listed in the Acknowledgments.
* msantin@bellvitgehospital.cat, msantin@ub.edu

## Abstract

### Background and purpose

Screening for and treatment of latent tuberculosis (TB) in patients with end-stage kidney disease (ESKD) are recommended. However, there is limited evidence on safety and treatment completion in this population. The objective of the study is to evaluate three short-course rifamycin-based regimens for the treatment of latent TB in ESKD patients.

### Methods

**Study design and setting**. This is a prospective, open label, randomized clinical trial, that will be conducted at seven teaching hospitals in Spain.

**Study population, randomization, and interventions**. Consecutive adult patients with ESKD requiring treatment for a latent TB infection will be randomly allocated (1:1:1) to receive one of the three treatment regimens of the study: three months of daily isoniazid plus rifampicin (3HR); three months of once-weekly isoniazid plus rifapentine (3HP); or four months of daily rifampicin (4R). Participants will be followed regularly through pre-established visits and a blood test schedule from enrolment to a month after finishing the assigned treatment.

**Data Availability Statement:** No datasets were generated or analysed during the current study. All relevant data from this study will be made available upon study completion.

**Funding:** This study is funded by the Instituto de Salud Carlos III through the grant "21/004444" (co-funded by the European Regional Development Fund/European Social Fund, "Investing in Your Future").

**Competing interests:** The authors declare that they have no conflict of interest.

**Outcomes**. The primary outcome will be treatment completion, while the secondary outcomes will be discontinuation of the assigned treatment due to adverse events, related or unrelated to the study treatment; definitive discontinuation of the assigned treatment because of adverse events related to the treatment of the study, and death.

**Sample size**. Two hundred and twenty-five subjects (75 per arm) will be enrolled, which will enable the demonstration, if it exists, of an increase of 0.16 in treatment completion rates either in the 3HP or 4R arm with respect to the 3HR arm.

## Discussion

Results of this clinical trial will contribute to evidence-based recommendations on the management of latent TB infection in ESKD patients.

## Trial registration

ClinicalTrials.gov identifier: NCT05021731.

## Introduction

The World Health Organization (WHO) regards the diagnosis and treatment of latent tuberculosis (TB) infection as an essential intervention to achieve the goal of the End TB Strategy by 2035 and the subsequent elimination of TB by 2050 [1].

End-stage kidney disease (ESKD) increases the risk of TB by a factor of 6 to 25, and the prevalence of TB among recipients of kidney transplants is 20- to 74-fold higher than that in the general population [2, 3]. Thus, systematic screening for and the treatment of TB infection in these patients are recommended [4]. However, ESKD patients constitute a fragile population, with frequent intercurrent events and polypharmacy, which pose major difficulties for treatment. Remarkably, despite the unanimity of the guidelines and the recommendations of the WHO, there is scarce evidence on treatment safety, tolerability, and completion in this particularly difficult-to-treat population [5–7].

The available evidence on the prevention of TB in ESKD patients in developed low-incidence countries is mostly based on treatment with isoniazid for 6–9 months [4]. In our experience with ESKD patients with TB infection, the vast majority of whom are treated with isoniazid-containing regimens prior to kidney transplantation, adverse events are the commonest cause of permanent discontinuation, with central nervous system toxicity being particularly significant [7]. Isoniazid-sparing or intermittent regimens have the potential advantages of shortening treatment duration, reducing toxicity, and improving preventive treatment completion. The combination of rifapentine and isoniazid administered weekly for 3 months (3HP) has been approved as a first-line treatment of latent TB infection in the USA, providing an appealing option to patients with advanced kidney disease [8–11]. However, although encouraging, the evidence so far is limited to a small number of cases [8]. Monotherapy with rifampicin for 4 months (4R) might be also a good option in terms of safety, as it has been shown to prevent active TB in a wide range of at-risk individuals in a clinical trial [12]. The 3-month combined isoniazid and rifampicin regimen (3HR) is one of the first-line regimens for the treatment of latent TB [13] and is widely used [7].

To date, no studies have prospectively evaluated the three short-course regimens (the daily 3HR, 4R, and weekly 3HP) for the treatment of latent TB in ESKD patients in a head-to-head

comparison. Based on available information, we hypothesize that either the daily rifampicin (4R) or the weekly isoniazid plus rifapentine (3HP) regimen will be better tolerated than the daily isoniazid plus rifampicin (3HR) regimen, resulting in higher treatment completion rates. We did not include the 6- to 9-month regimens of daily isoniazid as reference, as they are currently regarded as an alternative option and are gradually being replaced by the short rifampicin-containing regimens. In addition, the potential adverse effects of the prolonged administration of daily isoniazid can be captured in the 3HR regimen included in the trial.

## Materials and methods

### Aim, design and setting of the study

The study aims to determine if treatment completion with three months of once-weekly isoniazid plus rifapentine (3HP) or four months of daily rifampicin (4R) is better compared to that with three months of daily isoniazid plus rifampicin (3HR) for the treatment of latent TB infection in patients with ESKD. The study is an investigator-initiated, prospective, open-label, randomized clinical trial, which will be conducted at seven public hospitals in Spain, a country with a low incidence of TB.

### Sample size

The sample size was calculated to demonstrate clinically significant better completion rates with the 4R or 3HP regimen, respectively (experimental arms), compared to the 3HR regimen (control arm). With an one-side $\alpha = 0.025$, $\beta = 0.20$, and 5% expected losses, and assuming a 0.75 proportion of treatment completion in the control arm (3HR) according to our prior experience [7], 225 subjects (75 per arm) will be needed to demonstrate, if it exists, an increase in treatment completion rates of 0.16 in the respective experimental arms (4R or 3HP). The size of the sample was calculated with GIGAcalculator [14].

The 0.16 increase in compliance (from 0.75 to 0.93) was set according to the previous experience at the TB clinics of the coordinating centre. The rationale is that in our cohort of 534 ESKD patients treated in the last 9 years, central nervous system toxicity related to isoniazid occurred in more than 10% of patients, and it was the main reason for discontinuation of treatment. Therefore, assuming that such adverse event is avoided by using either once a week isoniazid (3HP regimen) or rifampicin alone (4R), together with the presumable reduction in liver toxicity with these two regimens, a compliance 0.93 is reasonable and achievable.

### Study population, allocation, and treatment arms

All consecutive adult patients with ESKD requiring treatment for latent TB will be eligible for the study. Inclusion criteria will be an age of 18 years or older, ESKD stage 5 (glomerular filtration rate [GFR] < 15 mL/min or under renal replacement therapy), and informed written consent to participate. Exclusion criteria will include prior allergy/intolerance to rifamycins or isoniazid, pregnancy or breastfeeding, pre-treatment transaminase (AST and/or ALT) levels > 5-fold higher than the upper limit of the normality, having received rifampicin or isoniazid within the two previous weeks, weight < 32 kg, taking drugs contraindicated with the study treatments or drugs that, although not contraindicated, interact with rifamycins and may seriously compromise patient health in the judgement of the investigators, and an inability to understand the study or provide written consent.

After signing informed consent, all eligible patients who fulfil the inclusion criteria will be randomly assigned to one of the three study arms: the control arm (3HR) or one of the experimental arms (3HP or 4R). Randomization will be stratified by the site at an allocation ratio of

1:1:1 using a computer-generated randomization list integrated into the Research Electronic Data Capture (REDCap) platform. None of the investigators will have access to the randomization list. The control arm (3HR) will consist of daily, self-administered, 5 mg/kg (up to 300 mg) of isoniazid plus daily 10 mg/kg of rifampicin (up to 600 mg) for three months. Experimental arm-1 (3HP) will consist of once-weekly, self-administered (with reminders via phone calls), 15 mg/kg (up to 900 mg) of isoniazid plus rifapentine (900 mg for those weighing $\geq$ 50 kg or 750 mg for those weighing 32–50 kg) for three months. Experimental arm-2 (4R) will consist of daily rifampicin (10 mg/kg) for four months. The 3HR and 3HP regimens will be supplemented with pyridoxine.

## Interventions

Follow-up of all participants will be carried out periodically at pre-specified time points by clinical interviews and blood tests (Fig 1).

Clinical interviews will be carried out with a structured questionnaire about adverse events, intercurrent clinical events, changes in medications, and treatment adherence. Blood tests will

| | STUDY PERIOD | | | | | | | |
|---|---|---|---|---|---|---|---|---|
| | Enrolment | Allocation | Post-allocation | | | | | Close-out |
| | Enrolment visit | Allocation visit | Visit 1 | Visit 2 | Visit 3 | Visit 4 | Visit 5 | Close-out visit |
| **TIMEPOINT, days** | **-14 to 0** | **0** | **14 ± 2** | **30 ± 2** | **60 ± 2** | **90 ± 2** | **120 ± 2** | **30 POST** |
| **Enrolment** | | | | | | | | |
| Eligibility screening | X | | | | | | | |
| Informed consent | X | | | | | | | |
| Allocation | | X | | | | | | |
| **Interventions** | | | | | | | | |
| Treatment with 3-mont HR (Control arm) | | | ← | | | → | | X |
| Treatment with 3-month HP (experimental-1 arm) | | | ← | | | → | | X |
| Treatment with 4-month R (experimental-2 arm) | | | ← | | | | → | X |
| **Assessments** | | | | | | | | |
| Demographics, medical history, physical exam, & chest X-ray [1] | X | | | | | | | |
| Blood test | X | | X | X | X | X [2] | X | |
| Concomitant medication | | | X | X | X | X | X [3] | |
| Adherence | | | X | X | X | X | X [3] | |
| Adverse events | | | X | X | X | X | X [3] | X |
| Treatment conclusion | | | | | | | | X |

POST= Post treatment; HR= Isoniazid plus rifampicin; HP= Isoniazid plus rifapentine; R= Rifampicin
[1] Chest X-ray or a CT scan available within the prior two months
[2] For the 3HR & 3HP arms
[3] For the 4R arm only

**Fig 1. Schedule of enrolment, interventions, and assessments.**

include at least blood cell counts and liver function tests (ALT, AST, alkaline phosphatase, and bilirubin levels). Adherence will be assessed by clinical interviews and the return of blister packs at visits. Participants included in the 3HP arm will be sent a reminder via text message or phone call to take their medication on the scheduled day. Adverse events will be graded according to the National Cancer Institute Common Toxicity Criteria Version 4.0 [15].

## Outcomes

The primary outcome will be treatment completion, defined as the completion of the treatment assigned for a predefined maximum accepted time (90 doses within 4 months without interruptions longer than 2 weeks and on no more than two occasions for the control arm [3HR]; 12 doses within a maximum of 14 weeks without interruptions longer than 10 days for experimental arm-1 [3HP]; and 120 doses within a maximum of 5 months without interruptions longer than 2 weeks and on no more than two occasions for experimental arm-2 [4R]). Secondary outcomes will be: definitive discontinuation of the assigned treatment because of adverse events, regardless of its relationship with the treatment of the study; definitive discontinuation of the assigned treatment because of adverse events related to the treatment of the study; and crude mortality, defined as deaths occurring during the study, regardless of its relationship with the study medication.

## Data management and statistical analysis

Data will be entered by the investigators at each site by means of an electronic CRF (eCRF) onto a database in the REDCap platform that is based at the Bellvitge Institute for Biomedical Research (IDIBELL). Efficacy analysis will be performed on the modified intention-to-treat population, which will consist of all randomized subjects. The proportions of participants who have completed the assigned treatment in the control arm (3HR) will be compared with those in either experimental arm-1 (3HP) or experimental arm-2 (4R). The analysis will be performed for the population as a whole and by subgroups (non-dialysis, haemodialysis or peritoneal dialysis). For safety, analyses will be performed on all randomized subjects who have received at least one dose of the assigned treatment. Likewise, proportions of permanent discontinuation of the assigned treatment, related and unrelated to the study medication, and death in the control arm will be compared with those of either of the experimental arms.

All statistical analyses will be performed by two investigators of the Biostatistics Unit of the IDIBELL, who will remain unaware of the trial-group assignments until the analyses have been completed. Differences between the arms will be assessed with Student's t-test if normally distributed or a two-samples Wilcoxon test if not normally distributed. Chi-square test will be used for categorical variables. Whenever possible, estimators will be given a 95% confidence interval (CI). Statistical significance will be set at a p-value < 0.025. Treatment of the data and analysis will be performed with the statistical package R version 3.4.1 or higher.

## Safety considerations

The active principles of the medications administered in the trial are licensed for the treatment of active and latent TB infection in the European Union (isoniazid and rifampicin) and USA (isoniazid, rifampicin, and rifapentine). Their safety profile is well established. However, because of the limited experience of administering a combination of rifapentine plus isoniazid in patients with ESKD, the protocol includes a frequent regular follow-up with visits and blood tests. Furthermore, the participants will have direct access to their treating team.

## Ethics

The sponsor and investigators are committed to conducting the study in accordance with the principles of the Declaration of Helsinki, ICH Guidelines for Good Clinical Practice and RD 1090/2015, fully conforming with the relevant regulations. Informed written consent will be obtained from all the participants, complying with the current legislation (Article 7 of RD 223/2004) and ethical principles of the Declaration of Helsinki. Treatment, communication, and cession of personal data of the participants will be guided by the "Ley Orgánica de Protección de Datos 15/99". The study was approved by the Ethics Committee of the Bellvitge University Hospital-IDIBELL (Approval number: PR281/21).

## Study timeline

The trial is planned to be carried out between 1 January 2022 and 31 December 2024, the period covered by the funding granted by the Instituto de Salud Carlos III and the European Regional Development Fund/European Social Fund. At the "pre-enrolment stage" (3 months), all the arrangements needed to start the trial will be completed, such as the procurement of informed consent, the development of standard operating procedures (SOPs), database design, and obtention of approval from the Competent National Authority (CNA). Participants will be enrolled during a 28-month period. Each participant will be followed for safety reasons for one month after finishing the assigned treatment. The final analysis of the data will be started one month after the last participant has completed their assigned treatment. A final report will be submitted to the funding bodies (usually within 3 months after finishing the study) and the CNA. Detailed information of the timeline of the trial is provided in the protocol.

## Discussion

Evidence-based guidance for the treatment of latent TB in ESKD patients is currently lacking. Patients with advanced kidney disease and candidates for transplants are not or hardly represented in the main clinical trials evaluating regimens with rifamycins [12, 16, 17]. The widely accepted recommendation on the systematic treatment of latent TB in patients with ESKD is based on the high risk of active TB and the scarce evidence on treatment safety and completion.

Based on the experience of our pre-transplant protocol of TB prevention (7) and the data reported in the literature, we will focus on three short-course rifamycin-containing regimens, as they are being increasingly used over the 6- to 9-month regimens of monotherapy with isoniazid for the treatment of latent TB [13, 18]. We anticipate a better tolerance of the 3-month intermittent regimen of isoniazid plus rifapentine and the 4-month rifampicin regimen with respect to the 3-month daily regimen of isoniazid plus rifampicin.

During the study, the protocol may need changes because of new information on efficacy and safety related to the drugs and regimens used in the trial. After discussion between the sponsor and the principal investigators at the different sites, amendments to the study will be submitted by the sponsor to the ethics committees and the CNA for approval. Likewise, termination of the trial may be considered because of evidence of superiority of the experimental regimens derived from the intermediate analysis or because of safety concerns. Decision on the termination of the trial will be proposed to the ethics committees and the CNA after assessment by an independent advice board.

The results of the trial will be made public only after the trial is closed. We expect to publish the results in a first-quartile journal of infectious diseases, transplantation or internal medicine, regardless of whether they are favourable or not.

The main limitation of the study is the open-label design that could influence the decisions on the continuation or discontinuation of the treatment assigned, which in turn could bias the achievement of the primary endpoint. Unfortunately, blinding treatment regimens by using placebo is not feasible due to the differences in the dosing frequency and duration of the three regimens.

In conclusion, knowledge on which treatment regimens are better tolerated and more convenient in this fragile population is needed. Hopefully, the results of this trial will contribute to the development of evidence-based recommendations on the management of latent TB infections in ESKD patients.

## Supporting information

**S1 File. Study protocol.**
(DOCX)

**S1 Checklist. SPIRIT 2013 checklist: Recommended items to address in a clinical trial protocol and related documents**[*].
(DOC)

## Acknowledgments

### The RIFAKiD-TB Trial team

- Miguel Santin[*]. Tuberculosis Unit, Service of Infectious Diseases, Bellvitge University Hospital-Bellvitge Institute for Biomedical Research (IDIBELL), University of Barcelona, L'Hospitalet de Llobregat, Barcelona, Spain.

[*]Head of the group. Email address: msantin@bellvitgehospital.cat; msantin@ub.edu

- Sandra Pérez-Recio. Tuberculosis Unit, Service of Infectious Diseases, Bellvitge University Hospital-Bellvitge Institute for Biomedical Research (IDIBELL), L'Hospitalet de Llobregat, Barcelona, Spain.

- Maria D. Grijota. Tuberculosis Unit, Service of Infectious Diseases, Bellvitge University Hospital-Bellvitge Institute for Biomedical Research (IDIBELL), University of Barcelona, L'Hospitalet de Llobregat, Barcelona, Spain.

- Núria Sabè. Tuberculosis Unit, Service of Infectious Diseases, Bellvitge University Hospital-Bellvitge Institute for Biomedical Research (IDIBELL), University of Barcelona, L'Hospitalet de Llobregat, Barcelona, Spain.

- Diego Sandoval. Department of Nephrology, Bellvitge University Hospital-Bellvitge Institute for Biomedical Research (IDIBELL), L'Hospitalet de Llobregat, Barcelona, Spain.

- Luís Anibarro. Tuberculosis Unit, Department of Internal Medicine, Complexo Hospitalario Universitario de Pontevedra, Pontevedra, Spain

- Olaia Conde. Tuberculosis Unit, Department of Internal Medicine, Complexo Hospitalario Universitario de Pontevedra, Pontevedra, Spain.

- Maria L. de Souza-Galvao. Servei de Pneumologia, Hospital Universitari Vall d'Hebron, Barcelona, Spain

- Nuria Saborit. Servei de Pneumologia, Hospital Universitari Vall d'Hebron, Barcelona, Spain

- Francisca Sánchez. Unitat Clínica de Tuberculosis, Hospital del Mar, Barcelona, Spain.

- Neus Jové. Unitat Clínica de Tuberculosis, Hospital del Mar, Barcelona, Spain.

- Rafael Luque. Clinical Unit of Infectious Diseases, Microbiology and Preventive Medicine Infectious Diseases Research Group, University of Seville/CSIC/University Virgen del Rocío and Virgen Macarena (IBIS), Sevilla, Spain.

- Maria D. Navarro. Infectious Diseases Department, University Hospital Virgen del Rocío, Seville, Spain.

- Paloma Gijón. Servicio de Microbiología Clínica y Enfermedades Infecciosas, Hospital General Universitario Gregorio Marañón. Instituto de Investigación Sanitaria Gregorio Marañón, Madrid, Spain.

- David Arroyo. Department of Nephrology, Hospital General Universitario Gregorio Marañón, Madrid, Spain.

- Clara Lucas. Servicio de Microbiología Clínica y Enfermedades Infecciosas, Hospital General Universitario Gregorio Marañón, Madrid, Spain.

- José M. Barcala. Infectious Diseases Unit, Hospital Universitario de Jerez, Jerez de la Frontera, Cádiz, Spain.

- Mercedes Garcia. Infectious Diseases Unit, Hospital Universitario de Jerez, Jerez de la Frontera, Cádiz, Spain.

- Cristina Ruiz. Service of Nephrology, Hospital Universitario de Jerez, Jerez de la Frontera, Cádiz, Spain.

## Author Contributions

**Conceptualization:** Miguel Santin, Sandra Perez-Recio, Maria D. Grijota, Luis Anibarro.

**Funding acquisition:** Miguel Santin.

**Methodology:** Miguel Santin, Luis Anibarro, Jose M. Barcala, Maria L. De Souza-Galvao, Paloma Gijon, Rafael Luque, Francesca Sanchez.

**Project administration:** Miguel Santin.

**Resources:** Maria D. Grijota.

**Writing – original draft:** Miguel Santin.

**Writing – review & editing:** Miguel Santin, Sandra Perez-Recio, Maria D. Grijota, Luis Anibarro, Jose M. Barcala, Maria L. De Souza-Galvao, Paloma Gijon, Rafael Luque, Francesca Sanchez.

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
