## [Decision Letter · Decision Letter 0]

13 Jul 2022

PONE-D-22-07891Comparison of three short-course rifamycin-based regimens for the prevention of tuberculosis in kidney transplant recipients: Study protocol for a randomised clinical trial (RIFAKiD-TB trial)PLOS ONE

Dear Dr. Santin,

Thank you for submitting your manuscript to PLOS ONE. After careful consideration, we feel that it has merit but does not fully meet PLOS ONE’s publication criteria as it currently stands. Therefore, we invite you to submit a revised version of the manuscript that addresses the points raised during the review process.

ACADEMIC EDITOR: Please revise the manuscript according to Reviewers' comments.

We look forward to receiving your revised manuscript.

Kind regards,

Justyna Gołębiewska

Academic Editor

PLOS ONE

Journal Requirements:

2. One of the noted authors is a group or consortium [insert name of group or team]. In addition to naming the author group, please list the individual authors and affiliations within this group in the acknowledgments section of your manuscript. Please also indicate clearly a lead author for this group along with a contact email address.’ 

4. We note that the original protocol file you uploaded contains a confidentiality notice indicating that the protocol may not be shared publicly or be published. Please note, however, that the PLOS Editorial Policy requires that the original protocol be published alongside your manuscript in the event of acceptance. Please note that should your paper be accepted, all content including the protocol will be published under the Creative Commons Attribution (CC BY) 4.0 license, which means that it will be freely available online, and any third party is permitted to access, download, copy, distribute, and use these materials in any way, even commercially, with proper attribution.

Therefore, we ask that you please seek permission from the study sponsor or body imposing the restriction on sharing this document to publish this protocol under CC BY 4.0 if your work is accepted. We kindly ask that you upload a formal statement signed by an institutional representative clarifying whether you will be able to comply with this policy. Additionally, please upload a clean copy of the protocol with the confidentiality notice (and any copyrighted institutional logos or signatures) removed.

Reviewers' comments:

Reviewer's Responses to Questions

**Comments to the Author**

1. Does the manuscript provide a valid rationale for the proposed study, with clearly identified and justified research questions?

Reviewer #1: Partly

Reviewer #2: Yes

Reviewer #3: Yes

2. Is the protocol technically sound and planned in a manner that will lead to a meaningful outcome and allow testing the stated hypotheses?

Reviewer #1: Partly

Reviewer #2: Yes

Reviewer #3: Yes

3. Is the methodology feasible and described in sufficient detail to allow the work to be replicable?

Reviewer #1: Yes

Reviewer #2: Yes

Reviewer #3: Yes

4. Have the authors described where all data underlying the findings will be made available when the study is complete?

Reviewer #1: No

Reviewer #2: No

Reviewer #3: Yes

5. Is the manuscript presented in an intelligible fashion and written in standard English?

Reviewer #1: Yes

Reviewer #2: Yes

Reviewer #3: Yes

6. Review Comments to the Author

You may also provide optional suggestions and comments to authors that they might find helpful in planning their study.

Reviewer #1: The authors present their protocol for a randomized controlled trial to assess the tolerability and treatment completion of 3 LTBI regimens in patients with end stage renal disease: control arm of 3HR (INH + Rifampin) daily for 3 months, experimental arm 1 = 3HP (INH + Rifapentine) once weekly for 3 months, and experimental arm 2 = 4R (Rifampicin) daily for 4 months. Overall the manuscript is clear and well-written.

A few comments and suggestions:

- In the original clinical trials: Three Months of Rifapentine and Isoniazid for Latent Tuberculosis Infection ( N Engl J Med 2011; 365:2155-2166 DOI: 10.1056/NEJMoa1104875), 3HP was compared to 9 months of INH and found that completion rate 13.1% higher in the 3HP group. Four Months of Rifampin or Nine Months of Isoniazid for Latent Tuberculosis in Adults (N Engl J Med 2018; 379:440-453 DOI: 10.1056/NEJMoa1714283), 4R was compared to 9 months of INH and found that completion rate was 15.1% higher in the 4R group. The authors of this protocol, propose to find a difference of 16% completion between the shorter regimens, which is likely an overestimation of the expected difference given the similarity of the regimens. The authors need to elaborate as to why they would expect a completion difference between 3HR and either of the (quite similar) comparison regimens that is larger than the differences observed in these large trials that included regimens of very different durations, or they may consider powering the study to detect a smaller percent completion difference between the groups.

- The authors need to provide rationale as to why Rifampin and Rifapentine have different exclusion criteria for drug-drug interactions (DDIs) as these two medications have very similar P450 interaction profiles

- Given the many DDIs with Rifamycins, a very important question in the patient population that the authors may consider investigating is: how many patients who are screened for enrollment are excluded based on DDIs. This will help shed some light on the polypharmacy and implications in this patient population.

- The authors need to provide rationale as to why self-administered therapy vs phone reminders are being used for the different arms. One study that can be cited for the use of self administered therapy for 3HP is the iAdhere trial (https://doi.org/10.7326/M17-1150)

- Quality of life measures comparing the regimens should be included to provide some patient-centered outcomes/information that may help inform clinician practice and selection of regimens, i.e. do patients who receive once weekly vs daily dosing report better tolerance or better quality of life.

- The authors need to provide rationale as to how and why the definitions for adherence and completion were selected as the current definitions do not appear to be in line with the original clinical trials that studied these regimens. For example, in PREVENT TB completion was counted if 11 doses of INH/rifapentine were completed within a 16-week period

- The title of the study focuses on prevention of TB in kidney transplant recipients but the study population is all patients with end-stage renal disease. Given prior work suggesting that latent TB treatment is not cost-effective in the general dialysis population because of poor survival (JAMA Intern Med. 2017;177(12):1755–1764. doi:10.1001/jamainternmed.2017.3941), the authors should justify why the regimens are being studied in a general population of patients with end-stage renal disease as opposed to only patients who might be eligible for kidney transplant (which is a presumably healthier subset and who may be able to tolerate these medications better)

Reviewer #2: I think the rationale for using a 0.025 alpha should be stated. Is the test one-sided 0.025 alpha (hence two-sided 0.05 alpha) or two-sided 0.025 alpha?

I think an active TB outcome measure in addition to treatment completion and death should be considered.

I am unclear how trial data will be made available after the study. The data availability part of the manuscript seems to say identifiable data will be made available to anyone. It might be better to upload de-identified trial data to a repository.

Reviewer #3: I have no meaningful comments to strengthen the paper. I see no real reason to publish a protoocol but the proposed protocol outline is sufficient.

7. PLOS authors have the option to publish the peer review history of their article (what does this mean?). If published, this will include your full peer review and any attached files.

Reviewer #1: No

Reviewer #2: No

Reviewer #3: **Yes: **Michael G. Ison

---

## [Author Response · Author response to Decision Letter 0]

1 Sep 2022

Reviewer #1: The authors present their protocol for a randomized controlled trial to assess the tolerability and treatment completion of 3 LTBI regimens in patients with end stage renal disease: control arm of 3HR (INH + Rifampin) daily for 3 months, experimental arm 1 = 3HP (INH + Rifapentine) once weekly for 3 months, and experimental arm 2 = 4R (Rifampicin) daily for 4 months. Overall, the manuscript is clear and well-written.

A few comments and suggestions:

- In the original clinical trials: Three Months of Rifapentine and Isoniazid for Latent Tuberculosis Infection (N Engl J Med 2011; 365:2155-2166 DOI: 10.1056/NEJMoa1104875), 3HP was compared to 9 months of INH and found that completion rate 13.1% higher in the 3HP group. Four Months of Rifampin or Nine Months of Isoniazid for Latent Tuberculosis in Adults (N Engl J Med 2018; 379:440-453 DOI: 10.1056/NEJMoa1714283), 4R was compared to 9 months of INH and found that completion rate was 15.1% higher in the 4R group. The authors of this protocol, propose to find a difference of 16% completion between the shorter regimens, which is likely an overestimation of the expected difference given the similarity of the regimens. The authors need to elaborate as to why they would expect a completion difference between 3HR and either of the (quite similar) comparison regimens that is larger than the differences observed in these large trials that included regimens of very different durations, or they may consider powering the study to detect a smaller percent completion difference between the groups.

Response: Thanks to the reviewer for they insightful comment. Certainly, according to the references mentioned, a difference of 16%, might not be expected. However, there are two very important points that should not be missed: First, the 2 studies mentioned did not include patients with end stage kidney disease (ESKD) (N Engl J Med 2011; 365:2155-2166), or very few (we do not know how many among 195 with immunosuppressive conditions in the study by Menzies in the N Engl J Med 2018; 379:440-453); and second, our own experience, on which we mainly based the sample size estimation, does not necessarily correspond to these two studies. 

Patients with ESKD represents a fragile population, have frequent intercurrent events, and polypharmacy, which often make LTBI treatment difficult. Therefore, results of the two previous studies cannot be extrapolated to this population. Unfortunately, evidence on the tolerance, safety, and compliance of treatment for LTBI in ESKD patients is scant. 

Because of lacking data in the field, our estimation was based on the scarce evidence available, data from our cohort of ESKD pre-transplant patients (Grijota MD. Transpl Infect Dis. 2021;00:e13603), and data from our general cohort of patients treated for LTBI. A retrospective, comparative study of 12-week RPT/INH and 9-month INH (Transplantation 2017;101: 1468–1472), showed higher treatment completion among the 43 patients in the 12-week RPT/INH (93%), compared to 47% among the 110 patients in the 9-month INH group. In our experience, more than 10% of ESKD patients treated with INH-containing regimens (3HR or 6-9H) presented CNS neurological toxicity, which resolved after discontinuing INH. CNS toxicity appeared, within the first week of treatment. Therefore, it is expected that the 3HR (control arm) will capture this frequent adverse event, and consequently an important decrease in the completion rates compared to the other two arms could be anticipated (obviously, 900 mg of INH, even given once a week, still may cause the same CNS toxicity!). Considering that 75% of ESKD patients in our cohort complete a course of treatment with the INITIAL INH-containing regimens (3HR or 6-9H), diminishing CNS and liver toxicity related to the daily INH, a compliance higher than 90% (rates in our cohort of pre-bio patients -data not published-), in one or both two arms with respect the 3HR (control arm) is achievable and realistic.

We have already pondered powering the study to detect smaller differences (this would not pose a difficulty as we have the potential of recruiting many more patients), as the reviewer suggest. However, differences lower than 15%-16% would not represent a substantial advantage of the alternative regimens with respect the 3HR one, and in our view, would not deserve to be explored. 

We have added a comment for further clarification (Page 8 of the manuscript). 

- The authors need to provide rationale as to why Rifampin and Rifapentine have different exclusion criteria for drug-drug interactions (DDIs) as these two medications have very similar P450 interaction profiles

Response: We agree with the reviewer’s comment on the interaction profile similarity between Rifampicin and Rifapentine. Since Rifapentine is not approved in Spain, we do not have available formal contraindications established by the Spanish Agency of Medicines and Medical Products (AEMPS) as we have for Rifampicin. According to the PRIFTIN datasheet in use in USA (This Medication Guide has been approved by the U.S. Food and Drug Administration. RIE-FPLR-SL-JUL21. Last revision: June 2020), the only contraindication for its use is hypersensitivity. Therefore, in consequence, and in accordance with the pragmatic approach of the trial, the concomitant use of other drugs with Rifapentine will be based on clinical judgement of treating clinicians, essentially in line with the recommendations for Rifampicin.

- Given the many DDIs with Rifamycins, a very important question in the patient population that the authors may consider investigating is: how many patients who are screened for enrollment are excluded based on DDIs. This will help shed some light on the polypharmacy and implications in this patient population.

Response: This is a very relevant question. In our practice at the TB clinics, a regimen including Rifampicin (3HR) is the first treatment option for LTBI, including ESKD patients. In our experience, we avoid the use of Rifamycins because of significative drug-drug interactions in 25% of ESKD patients (cohort of 534 patients treated at the TB clinics of the coordinator centre, in a 9-year period). However, although it is not easy to predict, we can expect a lower percentage of patients excluded in the context of a clinical trial. A log with patients and causes of exclusion and declining to participate will be part of information registered in the trial. 

- The authors need to provide rationale as to why self-administered therapy vs phone reminders are being used for the different arms. One study that can be cited for the use of self-administered therapy for 3HP is the iAdhere trial (https://doi.org/10.7326/M17-1150)

Response: The study the reviewer mentions is the only available evidence for the non-inferiority of self-administered (SAT) the 3HP regimen (designed with an excessively large margin of non-inferiority [15%]); and it would be applicable only in the USA. In Spain, among other countries, completion in the SAT-with-reminders groups was higher than in the SAT groups (84.8% vs.73.3%); but the study was not powered to evaluate this statistically. Since the trial is aimed at demonstrating higher rates of treatment compliance with this regimen, the investigator team considered the remainders a necessary intervention. A weekly phone remainder is feasible in our setting as the TB clinics participating in the study are staffed with specialist nurses with expertise in treatment of LTBI in different populations.

- Quality of life measures comparing the regimens should be included to provide some patient-centered outcomes/information that may help inform clinician practice and selection of regimens, i.e. do patients who receive once weekly vs daily dosing report better tolerance or better quality of life.

Response: We agree with the reviewer on the convenience of including a kind of quality-of-life measurement. Unfortunately, the budget of the study is already closed, and at this point this is not feasible. Ethics committee approved the clinical trial as it now stands.

- The authors need to provide rationale as to how and why the definitions for adherence and completion were selected as the current definitions do not appear to be in line with the original clinical trials that studied these regimens. For example, in PREVENT TB completion was counted if 11 doses of INH/rifapentine were completed within a 16-week period

Response: Overall, criteria for treatment compliance are tighter than criteria established in previous studies. The investigator team considers our criteria more appropriate for the population of the study, quite different to the participants profile in the mentioned studies. 

- The title of the study focuses on prevention of TB in kidney transplant recipients but the study population is all patients with end-stage renal disease. Given prior work suggesting that latent TB treatment is not cost-effective in the general dialysis population because of poor survival (JAMA Intern Med. 2017;177(12):1755–1764. doi:10.1001/jamainternmed.2017.3941), the authors should justify why the regimens are being studied in a general population of patients with end-stage renal disease as opposed to only patients who might be eligible for kidney transplant (which is a presumably healthier subset and who may be able to tolerate these medications better)

Response: We agree with the reviewer that the interest population is that of patients considered for transplant. Since in Spain, patients evaluated for and treated for LTBI are almost exclusively those who are evaluated for transplant, the population of the study will essentially be composed of pre-transplant patients. That is the reason why in the manuscript we assimilated “end-stage kidney disease” with “pre-transplant” population. 

We have changed the title of the manuscript. We have also corrected it throughout the manuscript.

Reviewer #2: 

-I think the rationale for using a 0.025 alpha should be stated. Is the test one-sided 0.025 alpha (hence two-sided 0.05 alpha) or two-sided 0.025 alpha?

Response: We have stated in the manuscript that it is one-side 0.025. Thanks for the advice. 

-I think an active TB outcome measure in addition to treatment completion and death should be considered.

Response: Thanks to the reviewer for this suggestion. Since the follow-up only spans to the time on treatment, it is a too short period of observation to active TB to occur. However, it makes sense to introduce a minor amendment to the protocol once the study is ongoing. 

-I am unclear how trial data will be made available after the study. The data availability part of the manuscript seems to say identifiable data will be made available to anyone. It might be better to upload de-identified trial data to a repository.

Response: We have introduced how data underlying findings and results will be available without restrictions (Page 3 of the manuscript). 

Reviewer #3: I have no meaningful comments to strengthen the paper. I see no real reason to publish a protocol but the proposed protocol outline is sufficient.

Response: Thank you!

---

## [Decision Letter · Decision Letter 1]

6 Oct 2022

Comparison of three short-course rifamycin-based regimens for the prevention of tuberculosis in patients with end-stage kidney disease: Study protocol for a randomised clinical trial (RIFAKiD-TB trial)

PONE-D-22-07891R1

Dear Dr. Santin,

We’re pleased to inform you that your manuscript has been judged scientifically suitable for publication and will be formally accepted for publication once it meets all outstanding technical requirements.

Kind regards,

Justyna Gołębiewska

Academic Editor

PLOS ONE

Additional Editor Comments (optional):

Reviewers' comments:

Reviewer's Responses to Questions

**Comments to the Author**

1. Does the manuscript provide a valid rationale for the proposed study, with clearly identified and justified research questions?

Reviewer #3: Yes

2. Is the protocol technically sound and planned in a manner that will lead to a meaningful outcome and allow testing the stated hypotheses?

Reviewer #3: Yes

3. Is the methodology feasible and described in sufficient detail to allow the work to be replicable?

Reviewer #3: Yes

4. Have the authors described where all data underlying the findings will be made available when the study is complete?

Reviewer #3: Yes

5. Is the manuscript presented in an intelligible fashion and written in standard English?

Reviewer #3: Yes

6. Review Comments to the Author

You may also provide optional suggestions and comments to authors that they might find helpful in planning their study.

Reviewer #3: I have no further comments to strengthen the manuscript. I have no further comments to strengthen the manuscript.

7. PLOS authors have the option to publish the peer review history of their article (what does this mean?). If published, this will include your full peer review and any attached files.

Reviewer #3: No

---

## [Editor Report · Acceptance letter]

13 Oct 2022

PONE-D-22-07891R1 

Comparison of three short-course rifamycin-based regimens for the prevention of tuberculosis in patients with end-stage kidney disease: Study protocol for a randomised clinical trial (RIFAKiD-TB trial) 

Dear Dr. Santin:

I'm pleased to inform you that your manuscript has been deemed suitable for publication in PLOS ONE. Congratulations! Your manuscript is now with our production department. 

Kind regards, 

on behalf of

Dr. Justyna Gołębiewska 

Academic Editor

PLOS ONE